# Baru Almond Beverage (*Baruccino*) with Different Sweeteners: Nutritional and Physical Properties and Exploration of Sensory and Non-Sensory Perceptions

**DOI:** 10.3390/foods15010127

**Published:** 2026-01-01

**Authors:** Laís Fernanda Batista, Raick Alves Ribeiro, Jucenir dos Santos Ferreira, Luana Cristina da Silva Ramos, Ana Clarissa dos Santos Pires, Márcia Cristina Teixeira Ribeiro Vidigal

**Affiliations:** 1Departamneto de Tecnologia de Alimentos, Universidade Federal de Viçosa, Viçosa 36570-900, Brazil; lais.batista@ufv.br (L.F.B.); jucenir.santos@ufv.br (J.d.S.F.); luana.c.ramos@ufv.br (L.C.d.S.R.); 2Faculdade de Farmácia, Universidade Federal da Bahia, Salvador 40170-115, Brazil; ribeiroraick@gmail.com

**Keywords:** allulose, sucrose, sweet taste, rheology, stability, sensory analysis, not milk

## Abstract

Winter beverage formulations made from baru almonds with the addition of sucrose (FS) or allulose (FA) were evaluated in two acceptance sensory sessions (blind test and test with information about the ingredients, their benefits, and the product label) by 100 potential consumers. The beverages were characterized for their proximate composition, pH, soluble solids content, instrumental color, microbiological analyses, steady state rheology, and kinetic stability test. The sensory acceptability of both formulations increased with information provision, reaching an acceptability index exceeding 70%. Furthermore, the sweet taste was the primary driver of acceptance, with consumers generally preferring sucrose as the sweetener. The perception of the sweet taste of FA compared to FS was 67% and 63% in the blind and informed tests, respectively, and showed similar results for physical, chemical, and rheological properties, as well as physical and microbiological stability. Thus, this study underscores the potential of allulose to replace sucrose without significantly altering product characteristics, showcasing its importance from both technological and sensory perspectives. Additionally, the novel product developed with baru contributes to the diversification and valorization of this Brazilian fruit, as well as being a tasty alternative for consumers.

## 1. Introduction

Products made from plants have enormous growth potential, especially when developed from native and underexplored sources with potential health benefits, such as the baru almond. The baru almond originates from the baruzeiro tree (*Dipteryx alata* Vog.), native to the Brazilian Cerrado, and represents a promising ingredient for developing novel products and disseminating knowledge about this and other lesser-known sources, thereby inspiring new combinations and flavors [1,2,3]. Additionally, baru almonds are rich in vitamins, proteins, lipids, and other components that provide nutritional and functional benefits, reinforcing their relevance for healthiness and enabling the development of products with greater appeal to health-conscious consumers [3,4,5].

Despite these advantages, the success of plant-based products in the market hinges significantly on their sensory quality. This critical factor is influenced by various elements, including sensory profile, information provided to consumers, and individual perceptions [6,7].

Alongside the increasing demand for plant-based foods, more consumers are actively seeking healthier dietary choices. This trend aligns with WHO recommendations to reduce the consumption of sodium, saturated fat, and added sugar. Among sucrose substitutes, allulose stands out as a promising alternative, offering sweetness comparable to that of sucrose.

Allulose, considered a rare sugar, is derived from figs and certain dried fruits. It is not metabolized by the body, catering to the demand for products with reduced sugar content and low calories (0.4 cal/g) and GRAS status by the Food and Drug Administration (FDA) [8,9,10]. Studies have evaluated the use of allulose to replace sucrose in products, such as cookies [11,12] and cakes [13,14], but not in plant-based beverages.

The formulation of foods with reduced added sugar aligns with the new labeling and mandatory nutritional information rules proposed by various governments, including Brazil, Mexico, and Chile. Front-of-package nutrition labeling, such as traffic light/advisory labels, magnifying glass, Nutri-Score, and numerical nutrition information labels, aims to provide consumers with clearer information and facilitate the identification and selection of products based on their ingredients through the information provided on the labels [15,16,17].

Despite these advancements, this study represents a scientific contribution regarding plant-based winter beverages developed with baru almonds, particularly with respect to sugar substitution using allulose and the influence of this information disclosure on sensory perception. In this sense, the objective was to evaluate how label information on these beverages affects sweetness perception when sucrose is replaced with allulose using consumer sensory evaluation under blind and informed conditions. In addition, physicochemical and technological characterizations were conducted, including proximate composition, pH, color, soluble solids content, microbiology, rheology behavior, and kinetic destabilization analyses. In this regard, the research has the potential to provide valuable insights for the food and beverage industry as it seeks to meet the growing demand for healthier and more sustainable products aligned with consumer expectations.

## 2. Materials and Methods

Roasted baru almonds, sucrose, cocoa powder, instant coffee, and cinnamon powder were obtained from local markets in Viçosa-MG (20°45′14″ S/42°52′55″ W). Allulose was imported for Commercial Elmar LTDA (São Paulo, Brazil) (lot: 1083756), and guar gum and xanthan gum were sourced from Sigma-Aldrich (St. Louis, MO, USA).

### 2.1. Development of Baruccino Beverages

The winter beverage formulations were developed at the Innovation Laboratory of the Department of Food Technology at the Federal University of Viçosa. After preliminary testing, two beverage formulations were produced, one with sucrose (FS) and the other with allulose (FA), with the remaining ingredients kept constant. The formulation development process comprised three primary stages, as illustrated in Figure 1. In the initial stage, roasted baru almonds were soaked and refrigerated (8 °C) for 15 h to enhance beverage palatability by removing residual water-soluble tannins. Following this, the soaking water was discarded, and the nuts were rinsed. Subsequently, in the second stage, the nuts were processed with hot water, filtered, and then reprocessed with the remaining ingredients. Lastly, in the third stage, the ingredients were homogenized, subjected to pasteurization (80 °C by 15 min), cooled, and stored under refrigeration.

Table 1 provides details of the ingredients utilized in the formulation development, along with the respective quantities added.

### 2.2. Proximate Composition and Nutritional Information

To construct the nutritional information table, the protein content was determined from total nitrogen using the Kjeldahl method [18], with a conversion factor of 5.75. The lipid content was obtained by the Mojonnier ether extraction method [19,20]. Moisture and ash content were determined gravimetrically using the [21] methodologies, and total carbohydrate content was obtained by difference.

The nutritional information for the formulations was compiled in accordance with the technical guidelines outlined in Resolution [22] and Normative Instruction IN 75 [23], in conjunction with the findings of the proximate composition analysis. Values for total sugar and added sugar were calculated from the quantities of each ingredient added. For sodium determination, the beverages were digested according to the methodology described by Ferreira et al. [24]. Sodium concentrations were then measured using a flame photometer (Celm FC-280, São Paulo, Brazil). All analyses were performed in triplicate.

### 2.3. Chemical and Physical Analyses

The pH of each formulation was measured in triplicate by inserting the pH electrode (pHmetro Gehaka PG2000, São Paulo, Brazil) directly into the sample. The soluble solids content (SST) was determined using a digital refractometer (pocket refractometer Pal^−1^, Atago, Japan) with a scale graduated in degrees Brix (°Brix). Color properties of the samples were evaluated by measuring the L* (lightness, ranging from black to white), a* (chromaticity from green to red), and b* (chromaticity from blue to yellow) parameters using a calibrated Colorquest^®^ XE colorimeter (HunterLab, Reston, USA) with a 10° observation angle and D65 illuminant (average daily sunlight). From these parameters, the cylindrical coordinate h* (hue angle) was calculated using the equations below:(1)*h* = *arct* (*b**/*a**)
(2)$$C*=\sqrt{{{\left(a*\right)}^{2}+\left(b*\right)}^{2}}$$
(3)$$∆E\;*=\sqrt{({∆L*)}^{2}+{\left(∆a*\right)}^{2}+{∆b*)}^{2}}$$

### 2.4. Rheological Behavior—Rotational Tests (Flow Curves)

Rheological analyses of the plant-based winter beverages were conducted using a Thermo Haake MARS rotational rheometer (Thermo Electron Corp., Karlsruhe, Germany), equipped with a thermostatic bath (Phoenix 2C30P, Thermo Electron Corp., Karlsruhe, Germany). Measurements were taken in triplicate using a Z20Din stainless steel sensor, at 60 °C. Flow curves of the beverages were determined by applying a shear rate from 0.01 to 250 s^−1^ in three curves (ascending, descending, and ascending) of 150 s each, measuring the corresponding shear stress (σ). The Herschel–Bulkley rheological model was fitted to the experimental data, represented by Equation (4).
(4)$$\sigma =\sigma_{0}+K\cdot {\overset{.}{\gamma }}^{n}$$

In which σ is the shear stress (Pa), σ_0_ is the yield stress (Pa), K is the consistency index (Pa·s^n^), n is the flow behavior index (dimensionless), and γ is the shear rate (s^−1^). Apparent viscosity was assessed at a shear rate of 100 s^−1^ to compare treatments, as established for the evaluation of thickened beverages [25,26,27].

### 2.5. Kinetic Stability

The kinetic stability of the formulations was assessed using a Turbiscan Lab (Formulation Smart Scientific Analysis, Dijon, France), which use an infrared light source (wavelength 880 nm) to irradiate the sample and capture backscattering (BS) and/or transmission (T) signals. After processing, 20 mL of each formulation was placed in glass cells and evaluated at 60 °C for 3 h, with data collected at 5 min intervals (37 scans). This procedure provided the Turbiscan Stability Index (TSI) values and the kinetic stability profile of the samples over time. The signals acquired over time concerning the sample’s height are calculated according to the software.

Furthermore, the TSI value can be used to classify samples based on their kinetic stability and to assess their correlation with visual perception, as illustrated in Figure 2.

To assess the impact of sweetener type and added gums on the kinetic stability of the formulations, two controls were developed: C1 with no sweetener addition, and C2 with no sweetener and gum additions.

### 2.6. Microbiological Analyses

To ensure the safety and health of untrained assessors, baruccino formulations were evaluated according to microbiological criteria established by the Normative Instruction [28] and Resolution RDC 724 [29]. The evaluation focused on the presence/absence of *Salmonella* spp., using a rapid test on Petrifilm (3M), standard plate count using the surface plating method for coagulase-positive *Staphylococci*, filamentous fungi, and Enterobacteriaceae following methodologies proposed by the Association of Official Analytical Chemists [21]. The results were compared with the limits established by Brazilian legislation.

### 2.7. Sensory Analysis

The sensory evaluation of formulations FS and FA was conducted in two sessions: a blind test and a second session with product information presented (informed test), both by the same group of consumers. A total of 100 regular consumers of winter beverages based on coffee and milk were invited to participate.

In both sessions, samples were evaluated using the Rate-All-That-Apply (RATA) and sensory acceptance tests. In the RATA test, attributes were assessed using a 3-point intensity scale ranging from not applicable (0), low (1), medium (2), to high (3). The 14 sensory attributes were defined based on descriptive studies of winter beverages in the literature [30,31] and on the tasting of winter beverages with baru. The terms raised were brown color, brightness, cocoa aroma, nutty aroma, sweet taste, unpleasant flavor, nutty flavor, cocoa flavor, cinnamon flavor, pleasant flavor, fatty flavor, creamy, homogeneous texture, and presence of particles. These attributes were presented to assessors in a random order.

Sensory acceptance of FS and FA was evaluated using a 9-point hedonic scale, ranging from 1 = disliked extremely to 9 = liked extremely, regarding the attributes of appearance, texture, flavor, and overall impression. The Acceptability Index (AI%) for each formulation was calculated using Equation (5).
(5)$$IA\left(\%\right)=\left(\frac{A}{B}\right)\times 100$$ which A = average score obtained for the formulation and B = maximum score given for the formulation [32].

To comprehend the commercial viability of the beverages, the assessment of purchase intention was conducted, indicating the extent to which assessors would be inclined to purchase the beverage after tasting it. Typically, options range from 5 = “Definitely Would Buy” to 1 = “Definitely Would Not Buy”, with intermediate values representing varying degrees of willingness to purchase. The sensory analysis sessions were conducted after the project’s approval by the Ethics in Human Research Committee at the Federal University of Viçosa (UFV), substantiated by the opinion number 5.291.419.

#### 2.7.1. First Session (Blind Test)

In the first session, the assessors were informed solely that they were evaluating a winter beverage. Formulations FS and FA were served at 60 °C, presented in a monadic sequence and random order, identified with random three-digit numbers. Consumers were provided with a sensory evaluation form (Appendix A). Form used in sensory analysis sessions) and water for mouth cleansing between samples.

#### 2.7.2. Second Session (Informed Test)

As in to the first session, following the instructions for the sensory test procedure, assessors were presented with samples in a randomized order. They were given the same sensory evaluation form as used in Session 1, along with a representation of a label for each formulation (Appendix A). Labels presented during sample evaluation, in the second session (informed test) and a brief description of the samples, highlighting the benefits of consuming baru almonds. At the end of this session, participants answered questions to collect demographic data, information about their consumption habits, and prior knowledge of baru almonds. The interval between the two sessions was at least 3 days.

### 2.8. Statistical Analysis

The experiment was conducted using a completely randomized design with two formulations, one with allulose (FA) and the other with sucrose (FS), in two replications. The results of the proximate, physical, and technological characterizations were evaluated using one-way ANOVA.

Sensory data were evaluated in two ways. For the comparison between FS and FA in each session (blind or informed), an analysis of variance (ANOVA) was performed, with two sources of variation (treatments and assessor) for each attribute. For the comparison between the blind and informed sessions for each formulation, paired-sample *t*-tests were performed to evaluate the influence of information on consumers’ perception of attributes in the RATA and on sensory acceptability.

The results are expressed as mean ± standard deviation. Statistical analysis was performed using R software (version 4.3.1), and the significance level adopted was α = 0.05.

## 3. Results and Discussion

### 3.1. Proximate Composition and Nutritional Information

The proximate composition of FS and FA is presented in Table 2. This analysis is essential for determining the proportion of nutrients in 100 mL of the product, which is used for developing nutritional labeling tables.

Protein content is an important indicator of the nutritional value of the formulations. The values found classify the winter beverage with baru as a potential source of protein since legislation IN 75 [23] considered a protein source the product that contains approximately 5 g/100 mL of food. In a study conducted by Zakidou et al. [] cappuccinos made from soy and a combination of soy and coconut presented average protein contents of 3.12 ± 0.06 and 1.33 ± 0.04 g/100 mL, respectively. Pojary et al. [33] when comparing the protein content of four types of commercial cappuccinos with and without milk, an average content of 2.8 g/100 mL was found. Therefore, winter beverage formulations with baru almonds showed higher protein levels than those found in studies with other plant-based beverages and commercial cappuccinos.

The total lipid content was the same for FS and FA, 5 g/100 mL. This level indicates that the beverage is low in saturated fats. Among the ingredients, baru almonds were the main contributors to the lipid fraction, containing approximately 38.20% to 43.60% of lipids, predominantly of unsaturated fatty acids [34].

The average total carbohydrate content was 29 g/100 mL for FS and 24 g/100 mL for FA, representing the sum of metabolizable total carbohydrates and dietary fibers.

The new labeling regulations aim to establish clearer, more objective rules for better communication between consumers and food label information, facilitating more informed, conscious choices about dietary habits. Notably, the added sugar content is highlighted, with values exceeding the limit established by Brazilian legislation for liquid products (7.5 g of added sugar/100 mL) necessitating a magnifying glass symbol on the front label [23]. Thus, FS should feature a seal on the front label indicating that it is “high in added sugar”. On the other hand, the FA formulation is considered sugar-free (<1 g/100 mL) and complies with FDA standards. The “sugar-free” claim is relevant to consumers seeking to limit their sugar intake. Pérez-Rodríguez et al. [35] noted that marketed plant-based beverages exhibit highly variable sugar content, ranging from 0 to 14.5 g of sugars/100 mL, influenced by the quantity and type of vegetable material and the addition of sugars for sensory purposes.

Regarding sodium level, both beverage formulations averaged, these sodium levels fall below the critical limit established for the use of nutritional claims such as “very low in sodium” (maximum 40 mg of sodium in 50 mL of the product). This aspect can enhance the attractiveness of the baru almond winter beverage to consumers.

Similarly, for FS and FA, the use of a magnifying glass on the front panel with information about saturated fat content is not obligatory, as they are below the critical limit for mandatory use (≥3 g/100 mL). Therefore, the product has the potential to be considered healthy, given that excessive intake of sodium and saturated fat through the diet is generally associated with negative health effects such as high blood pressure, obesity, and diabetes [36,37,38].

Craig et al. [39] analyzed 51 single-serve plant-based beverages based on the nutritional label listed on the commercial packaging and found that two thirds of them had high sugar levels (more than 5 g of total sugars/serving: 10% of Daily Value—DV), 63% were low in fat (not more than 1 g of saturated fat/serving: 5% of the DV), and only 39% were low in sodium (not more than 115 mg sodium/serving: 5% of DV). Therefore, the winter beverage containing baru almonds, that one with allulose (FA), is interesting from a nutritional aspect for consumers, especially for populations with restrictions on sugar, saturated fat, and sodium consumption.

Regarding the caloric content of the beverages, FA presented 160.2 kcal/100 g and FS 181.4 kcal/100 g. Replacing sucrose with allulose resulted in reduction caloric intake by 11.7%. According to Hossain et al. [40], allulose is classified as a low-calorie sweetener (<0.2 kcal/g), offering 90% fewer calories than sucrose. Therefore, the replacing sugars in foods and beverages with allulose can reduce their calorie and sugar content, as well as their glycemic and insulinemic impact. These effects can aid in weight management and reduce cardiometabolic risk factors [8,41]. Furthermore, traditional winter beverages made with milk typically have a higher caloric value compared to the plant-based formulations developed. Costa Fernandes et al. [42] reported caloric values ranging from 283.01 to 302.88 kcal/100 g for cappuccino-flavored dulce de leche containing different levels of instant coffee and cocoa powder.

Thus, the baru almond-infused plant-based winter beverages aim to align with WHO guidelines for reducing sugar, salt, and saturated fat intake, as a means of preventing and controlling diseases and complex risk factors through the consumption of healthier products. Additionally, the nutritional profile of baru almonds, which contain oleic and linoleic acids, minerals, and antioxidants, may help delay the onset of conditions such as cardiovascular diseases, and diabetes [43,44].

### 3.2. Chemical and Physical Characterization

FS and FA exhibited pH values of 6.04 and 5.94 and total soluble solids (TSS) of 33.08 and 32.7, respectively. The FS and FA formulations did not differ from each other (*p* > 0.05) concerning the color parameters L*, a*, b*, h°, and C* (Table 3).

Figure 3 illustrates the visual aspect of the *plant-based* formulations produced, which resemble traditional winter beverages made with milk, such as cappuccino.

Brightness (L* coordinate) closer to zero on a scale, indicating the where the darker sample, while closer to one hundred, the lighter. Thus, the winter beverages with baru almonds exhibited a dark hue, which can be explained by the cocoa characteristic. Regarding hue, FS and FA had a mean value of 57.75, indicating that their color was browner (hue angle = 40–75) than yellow (hue angle = 90) or orange (hue angle = 45). As for Chroma (C*), which gives the intensity or quantity of a shade, plant-based winter beverages showed a low C* (22.1 on average) since a more intense or saturated color. Furthermore, the ΔE value between FS and FA (0.55) indicates that the substitution of sucrose for allulose does not promote a difference in color perceptible to the naked eye (0 < ∆E < 1: observer does not notice the difference) [45].

### 3.3. Rheological Behavior

All formulations exhibited non-Newtonian fluid behavior with yield stress. The Herschel–Bulkley rheological model fits the experimental data, with R^2^ > 0.9997. The rheological parameters obtained for the fitted models are presented in Table 4.

The apparent viscosity of the winter beverages with baru almonds as a function of shear rate at 60 °C is illustrated in Figure 4. A reduction in apparent viscosity with increasing applied shear rate, indicating a pseudoplastic behavior, can be observed. This is confirmed by the value of n < 1 (Table 3). Yao et al. [46] obtained a similar shear-thinning behavior when evaluating the rheological properties of plant-based made beverages from six different sources (soy, peanut, adlay, azuki bean, oats, and buckwheat).

The apparent viscosity at 100 s^−1^ (η_100_) did not differ significantly between formulations (*p* > 0.05), with an average of 0.29 Pa.s. This value is consistent with literature findings under similar temperature conditions for nut-based beverages (η_100_ = 0.252 Pa.s) [47]. Sweeteners play a crucial role in the viscosity of foods, alongside proteins and lipids. Peasura et al. [48] suggest that the type of sweetener affects not only the physical and chemical properties but also the rheological behavior of samples, making sucrose substitution even more challenging. However, that replacing sucrose did not affect the flow behavior of winter beverages with baru almonds, as indicated by similar values of η_100_, τ_0_, k, and n for FS and FA.

### 3.4. Kinetic Stability

The kinetic stability of the formulated beverages FS and FA and the controls C1 and C2 was observed over a period of 180 min, at 60 °C, from which the Turbiscan Stability Index (TSI) scale was obtained. TSI is a dimensionless number that results from all occurrences of destabilization phenomena in the sample and is measured by a noticeable change in the backscatter or transmission signal intensity over the height of the sample, during the analysis period. Figure 5 shows the initial physical stability of the formulations at 5, 20, and 180 min.

Initially (Figure 5a), all four formulations exhibit excellent visual stability classified as A+ (dark green), with no significant destabilization observed and low TSI values (<0.42). In Figure 5b, a reduction in the kinetic stability of plant-based winter beverages is observed at 20 min of scanning, especially for C1 (formulation without added sweeteners). The FA, C2, and FS were classified as A (light green), although the sample is visually good, indicating that the equipment begins to detect the presence of destabilization mechanism at a very early stage. However, in 90% of cases, such destabilization phenomena are not visible or are perceptible to the naked eye. C1 is classified as B (yellow) and presents a TSI value above 1.0, indicating the onset of some destabilization phenomenon, i.e., changes occurring in the sample height relative to the reference point. When the samples have TSI scale values between 1.0 and 3.0, destabilization becomes detectable. Cichońska et al. [49] used Turbiscan to evaluate the kinetic stability of bean-based beverages (non-germinated beans—BG and germinated beans—G). Beverages B and BG exhibited TSI values of approximately 0.6–0.9 and 1.3–2.0, respectively, indicating distinct stability profiles. These results further confirm the sensitivity of the technique for monitoring destabilization over time under refrigerated storage conditions.

Figure 5c indicates that after 180 min of analysis, winter beverages containing baru almonds underwent destabilizing. The observed reduction in kinetic stability is mainly due to the sedimentation of cocoa and soluble coffee particles, which settle to the bottom of the glass cell, leading to a macroscopic phase separation and consequent sample clarification. This undesirable characteristic may affect product acceptance. Ni et al. [50] also reported destabilization phenomena when assessing the physical stability of a cloudy ginkgo beverage, despite the incorporation of a combination of thickeners (microcrystalline cellulose and gellan gum) at different concentrations. Turbiscan analysis revealed irreversible phase separation, which may have been induced by particle size effects and by the reduction in viscosity during heat treatment.

The destabilization mechanisms associated with increased lipid diffusion may be related to both coalescence (particles within the system colliding and forming a new particle) and Ostwald ripening (smaller particles transferring through diffusion of their molecules to others, increasing in size and becoming part of them) [26]. Over time, at a temperature of 60 °C, lipid diffusion contributes to the occurrence of Ostwald ripening and coalescence, which over time may lead to macroscopic phase separation, as evidenced by changes detected in the analysis and cream formation at the top, where under the studied conditions, it occurred due to the increased speed at which particles rise, influenced by the average diameter and density, expressed by the Stokes Equation (6).
(6)$$vs=\frac{d^{2}\left(\rho_{p}-\rho_{L}\right)\;*\;g}{18_{\eta }}$$

Thus, over time, even though they have been homogenized, the new membrane formed is mainly due to the adsorption of proteins on the surface of lipids, and this may not have been completely effective in forming more uniform globules, despite the likely reduction in their size, which increases over time and causes the samples to lose their kinetic stability. These detected variations indicate that the destabilization mechanisms described were detected by the equipment after 180 min. The TSI values < 3.0 obtained indicate that the formulations remained visually stable, without perceptible visual alterations, thus demonstrating the Turbiscan’s ability to accurately detect small changes in the evaluated samples. In descending order of kinetic instability, C1 (TSI = 1.7) was the most unstable, followed by formulations FS (TSI = 1.5) and FA (TSI= 1.5), and C2 (TSI = 1.0), which was the most stable sample among them.

Figure 6 illustrates the overall destabilization kinetics of the plant-based winter beverage, showing variation in TSI values over the analysis time. The most significant increase in TSI occurs in the first 20 min of analysis and remains constant, except for C1, which had a more pronounced and variable increase than the other formulations (FS, FA, and C2), showing more uniform and homogeneous changes. Although the instability phenomena of the four formulations were noticeable in the first 20 min, under the test conditions, with destabilization mechanisms detected by the equipment over time, no visual changes were observed. Therefore, this does not compromise the product ’s appearance.

The higher kinetic instability observed in C1 suggests that solids play a crucial role in the overall stability of plant-based beverages. The elevated TSI values indicate that the sample tends to clarify over time due to phase separation, particularly in the upper and lower regions where the most significant changes occur. FS and FA exhibited similar behavior, with FS being marginally less stable than FA. This underscores the feasibility of allulose as a sucrose substitute without compromising the product’s technological properties. Consequently, winter beverages containing baru almonds with either sucrose (FS) or allulose (FA) demonstrate satisfactory kinetic stability at the consumption temperature (60 °C) for up to 180 min. Furthermore, consuming a portion of this type of product takes less time than the analysis time.

### 3.5. Microbiological Analyses

As sugar is recognized for its role in food preservation, its substitution can affect the food’s microbiological quality. Therefore, microbiological analyses were conducted to ensure the beverage’s fitness for consumption, and the results are presented in Table 5.

The FS and FA meet the microbiological criteria described in legislation, are considered acceptable quality with respect to filamentous fungi and *Enterobacteriaceae* counts and are free of *Salmonella*. Additionally, after confirmatory tests, negative results were obtained for coagulase-positive *Staphylococci.* According to Prado et al. [51], *Salmonella* spp., *Escherichia coli*, and *Staphylococcus* spp. are the principal causes of outbreaks associated with the consumption of contaminated foods, with handling and poor hygiene being the main factors responsible for contamination. Most of these outbreaks are linked to animal products, but they can also affect plant-based products that are mishandled. However, no outbreaks associated with them have been reported in the current literature [52,53,54].

These results are consistent with the values previously reported and described in the scientific literature for plant-based beverages made with almonds and other sources. Di Renzo [55], when microbiologically evaluating different plant-based beverages (almond, Brazil nut, macadamia, cashew nut, pistachio, and oat), reported that, in general, the formulations presented microbial counts within the limits required by legislation. These authors concluded that microbiological quality is a critical factor in ensuring the safety of these beverages and is related to the intrinsic composition of the product and the level of rigor adopted throughout the processing.

In this context, plant-based products generally demonstrate enhanced food safety from a microbiological perspective, attributed to the implementation of good manufacturing practices throughout processing, which is pivotal to product quality. Consequently, the formulations can be deemed safe for consumption during sensory evaluation, posing no risk to consumer health, and meeting acceptable quality standards as stipulated by Brazilian legislation.

### 3.6. Sensory Analysis

The panel of assessors participating in the study comprised 100 volunteers, 68% of whom were female and 32% male, with ages ranging between 18 and 65 years. Regarding the sensory characterization of the FS and FA beverages using RATA in both sessions (blind and informed tests), significant differences (*p* < 0.05) were found between the formulations for the attributes sweet taste, pleasant flavor, and unpleasant flavor.

The replacement of sucrose with allulose did not significantly affect the sensory perception of the attributes: brown color, brightness, cocoa aroma, nutty aroma, cocoa flavor, sweet taste, nutty flavor, cinnamon flavor, fatty, unpleasant flavor, creamy, pleasant flavor, homogeneous texture, and presence of particles (Table 6). Based on the scores obtained, the attributes of unpleasant flavor, cinnamon flavor, and fatty (blind test) do not characterize winter beverages with baru almonds, as the averages are below 1, ranging from “not applicable” to “low” perceived intensity. The pleasant flavor intensity for FS was 2.24 and 2.27 and for FA it was 1.61 and 1.77, in the blind and informed tests, respectively.

After receiving the label and information (session 2—Informed test), the values of the FS attributes increased from session 1 to session 2 and they were considered significantly different (*p* < 0.05) between sessions 1 and 2, for the attributes cocoa flavor, cinnamon flavor, presence of particles, homogeneous texture, and brightness. No significant differences (*p* > 0.05) were found for the other evaluated attributes. On the other hand, for FA, there was a significant difference between the blind and informed tests (*p* < 0.05) for the attributes cocoa aroma and cocoa flavor, with a reduction in the perceived intensity of these attributes and an increase in the perceived intensity for brightness and pleasant flavor. Therefore, the information about allulose had a significant positive effect on perception of pleasant flavor and brightness.

In the blind test, the greater intensity was attributed to FS (1.61), compared to FA (1.08), corresponding to a perceived sweetness equivalent to 67% of allulose relative to sucrose. In the informed test, the means for FS and FA were m = 1.54 and m = 0.97, respectively, indicating that allulose presented sweetness equivalent to 63% of sucrose. The sweetness equivalence results for the FA beverage in the blind test may be comparable to those reported in the literature, which describe allulose (70%) as equivalent in sweetness to [11,56]. In this case, the type of sweetener (sucrose or allulose) affected the perception of sweet taste equivalence, an important attribute in beverages acceptance [11,41,55]. Furthermore, consumers rated the intensity of the sweet as “low” for FA (means close to 1) and as “low” to “medium” for FS (means between 1 and 2), based on the results of both sessions.

The hedonic means for the attributes evaluated in the acceptance test and the acceptability index are presented in Table 5. In the blind test, FS and FA were statistically equal concerning the acceptance of appearance, aroma, and texture (*p* > 0.05). For flavor and overall impression, the formulation containing sucrose (FS) was preferred the one containing allulose (FA) (*p* < 0.05). FS presented hedonic means between 6.62 (texture) and 7.94 (appearance) in the blind test, and between 6.98 (texture) and 8.18 (appearance and aroma) in the informed test. The hedonic means for FA varied between 5.59 (flavor) and 7.93 (appearance) in the blind test and between 5.86 (flavor) and 8.03 (appearance) in the informed test; the means overall impression was between the hedonic terms “I liked slightly” and “I liked moderately”.

Information did not influence FA acceptability, as hedonic means did not differ significantly across sessions (*p* > 0.05). However, FS was significantly influenced by information (*p* < 0.05) on appearance, aroma, texture, and overall impression, with an increase in average acceptability across all attributes. The AI% values for FS formulation also increased in the test with information, exceeding 72.66%, except for the flavor attribute. For a product to be considered accepted, AI should be at least 70% based on sensory properties. Thus, both formulations were well accepted by general consumers with no dietary restrictions, who accounted for more than 90% of the participants. This result suggests that the plant-based winter drinks developed could be attractive not only to vegans, vegetarians, and flexitarians, but also to consumers who do not follow restrictive diets. Rincon et al. [57] developed a new beverage based on chickpea and coconut, with low sensory acceptability (<70%) among general consumers, but higher acceptability averages when the same samples were evaluated in a second session with consumers more specific to this type of beverage.

Therefore, FA has the potential to be better accepted by consumers, particularly through formulation adjustments to increase its sweetness, which may have compromised the flavor of the plant-based winter beverage. Furthermore, a few differences were observed between FS and FA regarding the other attributes evaluated, suggesting that allulose can serve as a good substitute for sucrose without compromising the product’s sensorial characteristics of appearance, texture, and aroma.

The results of the means obtained by the attitude scale, which aims to measure the potential purchase intention of the FS and FA beverages, are presented in Table 7. The information did not significantly affect (*p* > 0.05) the purchase intention for FS. However, higher mean purchase intention scores were observed for this formulation in both the blind and informed tests, with scores above 3, indicating a tendency to consider purchasing the FS. The information contained on the front label “high in added sugar” did not discourage consumers from purchasing the product.

For FA, significant differences were observed between sessions 1 and 2 (*p* < 0.05), indicating the positive impact of information on purchase intention. There was an increase in the mean attitude scale score, transitioning from “probably wouldn’t buy” to “might or might not buy” with a mean score of 2.88 in the blind test, to “might or might not buy” to “probably would buy” with a mean score of 3.22 in the informed test. Information regarding the nutritional table, absence of added sugar on the front label, and the health benefits of baru contributed to an enhanced purchase intention for the plant-based winter beverage sweetened with allulose. This result corroborates the study by Khan et al. [58], which evaluated the impact of marketing interventions on sugar-free and sugar-sweetened soft drink sales and reported that the interventions had a statistically significant impact on sugar-free drinks but not for sugar-sweetened drinks. On average, sales of the sugar-free product were 56.75% higher than those of the control site with added sucrose.

When asked about the habit of consuming sweetened winter beverages, 61% of consumers indicated consuming such products with added sugar, while 22% use sweeteners (natural or artificial), and only 17% do not add any sweetener, as illustrated in Figure 7. As a result, most of the untrained assessors indicated a habit of consuming and familiarity with products sweetened with sucrose. This statement, while advocating the exploration of alternative sweeteners to replace sucrose, highlights the need for viable substitutes that meet consumers’ preferences without compromising product quality [13,58,59].

According to Singh [60], the sweet taste can drive consumption of a specific product, shaping consumer behavior. These authors also propose a direct correlation between sweetness and the perception of a food’s non-toxicity, suggesting that sweeter tastes are perceived as less toxic, serving as a survival mechanism throughout history. Li et al. [6] assessed the preferences and perceptions of various consumer groups regarding the intake of different low-calorie sweeteners using a validated questionnaire. They found that older individuals tended to avoid sweeteners and reported that their consumption was perceived as more healthful than sucrose. Due to the similarities in the perception of some sensory attributes and sensory acceptability, we suggest that allulose can serve as a good substitute for sucrose without compromising the sensory quality, indicating differences in sweetness perception in formulations where FS was indicated as sweeter than FA.

Furthermore, it is suggested that information on the nutritional benefits of baru almonds may enhance the product’s acceptance, especially considering that only 25% of the participants were familiar with this type of nut.

It is believed that awareness of these benefits can drive the development and consumption of plant-based beverages made from baru almonds, as 50% of participants expressed interest in consuming such products. The nutritional value, particularly in terms of protein content, attributed to the inclusion of baru almonds, has the potential to enhance the acceptance of plant-based winter beverages. Presently, consumers are increasingly seeking information about the ingredients and components in food, contributing to improved health through the benefits they provide [61,62]. According to Batista et al. [63] implementing marketing strategies, such as reformulating labels and highlighting the presence of native ingredients, can meet consumers’ demands for plant-based foods. These authors propose three main themes to represent consumers’ perspectives on plant-based foods: sustainability and ethics; health and wellness; and sensoriality and pleasure.

In a systematic evaluation of the effects of nutritional claims on consumer behavior, Balco et al. [64] found that factors such as familiarity and awareness of the nutritional benefits associated with consumption can positively influence choices. Consequently, the new product offers a nutritious alternative with sensory quality to cater to various consumer groups, encompassing vegans, vegetarians, flexitarians, and individuals without dietary restrictions.

## 4. Conclusions

The replacement of sucrose with allulose can be carried out without modifying the proximate composition, the physicochemical composition, rheological behavior, and kinetic stability of the developed plant-based beverages, which expands the potential application of allulose in plant matrices. Additionally, the winter beverages containing baru almond with sucrose (FS) or allulose (FA) were well accepted by the untrained assessors. However, consumers showed greater acceptance of FS, characterized by greater sweetness intensity. The similarity in the perceptions of sensory attributes (except sweetness) indicates that allulose is a promising substitute for sucrose.

The search for sweeteners that replace sucrose is a challenge in terms of technology, stability, and sensorial quality. Therefore, allulose emerges as a potential alternative to replace sucrose in plant-based beverages, appealing to consumers seeking well-being and health. The winter beverage with baru almond, especially the one with allulose, has a good nutritional profile for the consumer due to its protein content (around 4.5%), low saturated fat, and being free of added sugar. Moreover, the status of the beverage as an excellent source of essential nutrients that offers associated health benefits makes it appealing to various consumer groups concerned with well-being and flavor. Besides the nutritional advantages, the winter beverage containing baru almonds can also contribute to sustainability via the responsible sourcing of ingredients, by promoting local, environmentally friendly production methods, and by adding value to products.

## Figures and Tables

**Figure 1 foods-15-00127-f001:**
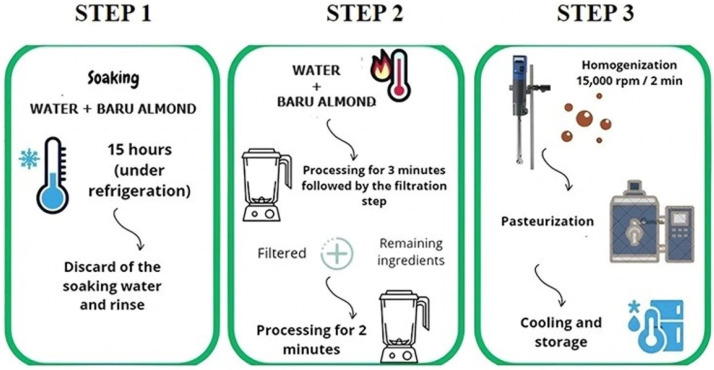
Stages of general processing used in the development of plant-based winter beverages.

**Figure 2 foods-15-00127-f002:**
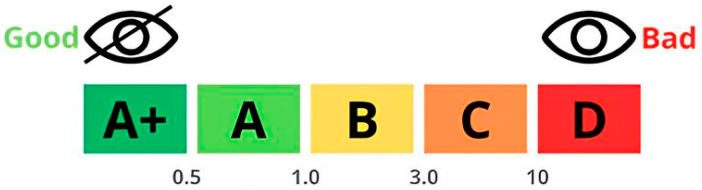
TSI rating scale according to stability.

**Figure 3 foods-15-00127-f003:**
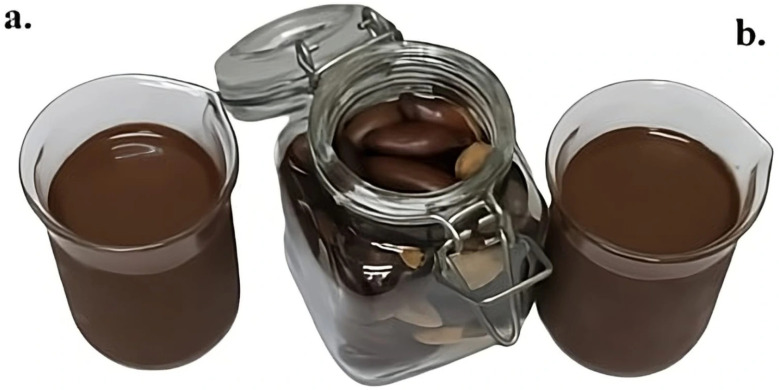
Winter beverages with baru almonds: (**a**) formulation with allulose (FA), (**b**). formulation with sucrose (FS).

**Figure 4 foods-15-00127-f004:**
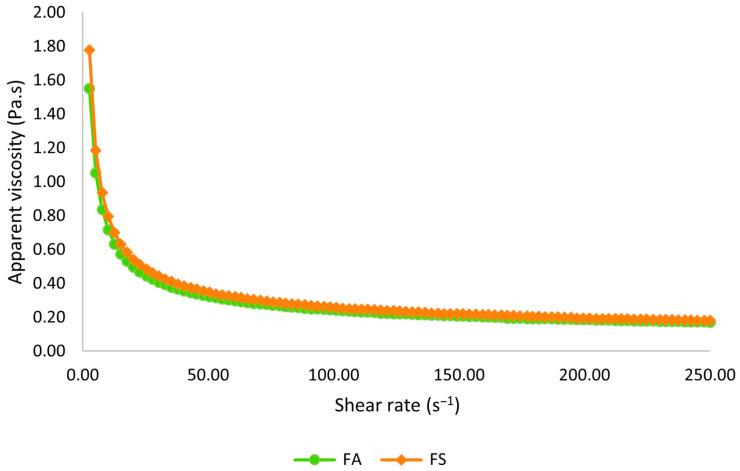
Apparent viscosity as a function of shear rate at 60 °C. FA = formulation with allulose; FS = formulation with sucrose.

**Figure 5 foods-15-00127-f005:**
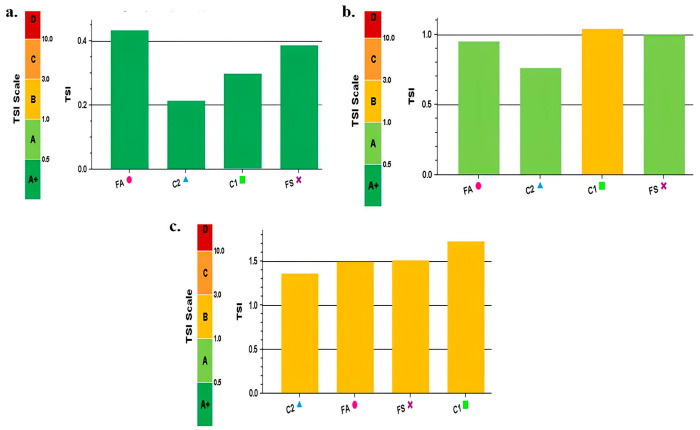
TSI Scale (**a**) 5 min scanning, (**b**) 20 min scanning of samples, (**c**) 180 min scanning evaluated at 60 °C. FA = formulation with allulose; FS = formulation with sucrose; C1 = formulation without addition of sweeteners; C2 = formulation without addition of sweeteners and gums.

**Figure 6 foods-15-00127-f006:**
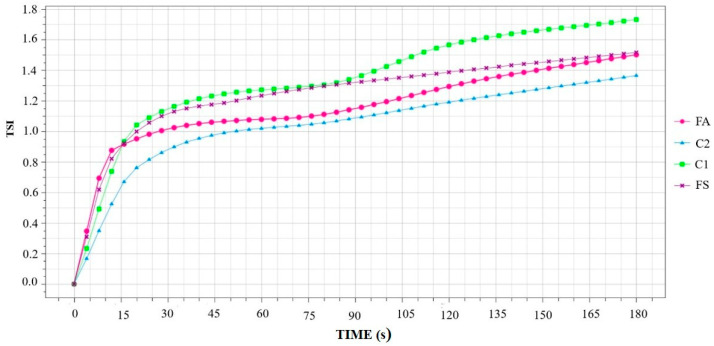
Global Destabilization Kinetics. FA = formulation with allulose; FS = formulation with sucrose; C1 = formulation without the addition of sweeteners; C2 = formulation without the addition of sweeteners and gums.

**Figure 7 foods-15-00127-f007:**
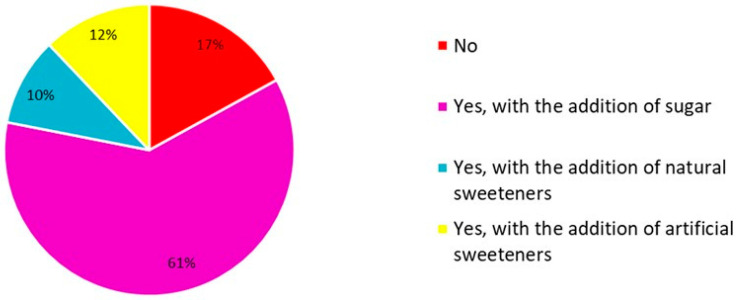
Percentage of response in relation to the consumption of sweetened winter beverages.

**Table 1 foods-15-00127-t001:** Ingredients used in the development of the plant-based winter beverages and their respective percentages.

Ingredients	Percentage (%)
Water	58.5
Baru nut	19.5
Sweetener *	14.8
Cocoa powder (100%)	5.1
Instant coffee	1.6
Cinnamon powder	0.2
Guar gum	0.2
Xanthan gum	0.1

* sucrose or allulose.

**Table 2 foods-15-00127-t002:** Composition proximate and nutritional of winter beverages with the almond baru.

Composition (g/100 mL)	FS	FA
Total water content	61.0 ^b^ ± 0.0	66.0 ^a^ ± 0.4
Carbohydrates	29.0	24.0
Proteins	4.4 ^a^ ± 0.2	4.6 ^a^ ± 0.1
Lipids	5.0 ^a^ ± 0.5	5.0 ^a^ ± 0.3
Ash	0.01 ^a^ ± 0.00	0.01 ^a^ ± 0.00
Sodium	40.0 ^a^ ± 2.0	40.0 ^a^ ± 3.0

Means followed by the same letter in the row do not differ from each other by ANOVA (*p* > 0.05). FA = formulation with allulose; FS = formulation with sucrose.

**Table 3 foods-15-00127-t003:** Instrumental color of beverages with baru almonds.

Color Properties	FS	FA
L*	24.6 ^a^	24.1 ^a^
a*	11.74 ^a^	11.84 ^a^
b*	18.57 ^a^	18.77 ^a^
h°	57.6 ^a^	57.9 ^a^
C*	22.0 ^a^	22.2 ^a^

Means followed by the same letter in the row do not differ from each other by ANOVA (*p* > 0.05). FA = formulation with allulose; FS = formulation with sucrose.

**Table 4 foods-15-00127-t004:** Rheological parameters by the Herschel–Bulkley model of the winter beverages with baru almond.

Rheological Properties	FS	FA
τ_0_ (Pa)	2.28 ^a^ ± 0.15	1.82 ^a^ ± 0.29
K (Pa·s^n^)	1.45 ^a^ ± 0.20	1.27 ^a^ ± 0.15
n	0.62 ^a^ ± 0.02	0.63 ^a^ ± 0.01
R^2^	0.9998 ± 0.0001	0.9999 ± 0.0001

σ_0_ is the yield stress, K is the consistency index, n is the flow behavior index. Means followed by the same letter in the row do not differ from each other by ANOVA (*p* > 0.05). FA = formulation with allulose; FS = formulation with sucrose.

**Table 5 foods-15-00127-t005:** Microbiological analyses of the winter beverages with baru almond.

Microbiological Analyses	FS	FA
Filamentous Fungi (CFU/mL)	1.6 × 10^2^	4.1 × 10^1^
Coagulase-positive Staphylococci (CFU/mL)	negative	negative
*Enterobacteriaceae* (CFU/mL)	<1/mL (est.) *	<1/mL (est.) *
*Salmonella* (Presence/Absence 25 mL)	Absence	Absence

* estimated.

**Table 6 foods-15-00127-t006:** Mean values of attributes obtained in the RATA test conducted in blind and informed conditions, for winter beverages with baru almond.

Descriptor Terms	FS (Blind Test)	FS (Informed Test)	FA (Blind Test)	FA (Informed Test)
Brown color	2.74 ^aA^	2.75 ^aA^	2.73 ^aA^	2.75 ^aA^
Brightness	2.41 ^aA^	2.50 ^aB^	2.48 ^aA^	2.57 ^aB^
Cocoa aroma	1.94 ^aA^	1.83 ^aA^	1.99 ^aA^	1.80 ^aB^
Nutty aroma	2.02 ^aA^	2.02 ^aA^	1.97 ^aA^	2.09 ^aA^
Sweet taste	1.61 ^aA^	1.54 ^aA^	1.08 ^bA^	0.97 ^bA^
Cocoa flavor	2.18 ^aA^	1.89 ^aB^	2.19 ^aA^	1.87 ^aB^
Pleasant flavor	2.14 ^aA^	2.27 ^aA^	1.61 ^bA^	1.77 ^bB^
Unpleasant flavor	0.70 ^aA^	0.60 ^aA^	0.98 ^bA^	0.93 ^bA^
Nutty flavor	2.25 ^aA^	2.20 ^aA^	2.16 ^aA^	2.21 ^aA^
Cinnamon flavor	0.83 ^aA^	0.86 ^aB^	0.81 ^aA^	0.93 ^aA^
Fatty	0.97 ^aA^	0.95 ^aA^	0.98 ^aA^	1.11 ^aA^
Creamy	1.74 ^aA^	2.16 ^aA^	1.64 ^aA^	2.02 ^aA^
Homogeneous texture	2.13 ^aA^	2.32 ^aB^	2.11 ^aA^	2.24 ^aA^
Presence of particles	1.36 ^aA^	1.10 ^aA^	1.27 ^aA^	1.17 ^aA^

Equal lowercase letters on the same row indicate no significant difference between FS and FA within each session (blind and informed), according to ANOVA (*p* > 0.05). Equal uppercase letters indicate no significant difference between blind and informed tests for each formulation, according to paired-samples *t*-test (*p* > 0.05). FA = formulation with allulose; FS = formulation with sucrose.

**Table 7 foods-15-00127-t007:** Hedonic means, acceptability index (AI, %) and purchase intention for the winter beverages with baru almond.

Sensory Measures	FS (Blind Test)	FS (Informed Test)	FA (Blind Test)	FA (Informed Test)
Appearance	7.94 ^aA^	8.18 ^aB^	7.93 ^aA^	8.03 ^bA^
AI	88.22	90.88	88.11	89.22
Aroma	7.12 ^aA^	8.18 ^aB^	6.91 ^aA^	7.10 ^bA^
AI	79.11	90.88	76.77	78.88
Texture	6.62 ^aA^	6.98 ^aB^	6.36 ^aA^	6.58 ^bA^
AI	73.55	77.55	70.66	73.11
Flavor	6.65 ^aA^	6.99 ^aA^	5.59 ^bA^	5.86 ^bA^
AI	73.88	76.66	62.11	65.11
Overall impression	6.93 ^aA^	7.33 ^aB^	6.37 ^bA^	6.54 ^bA^
AI	76.99	81.44	70.70	72.66
Purchase intention	3.38 ^aA^	3.61 ^aA^	2.87 ^bB^	3.22 ^bA^

Equal lowercase letters, on the same row indicate no significant difference between FS and FA within each session (blind or informed), based on ANOVA (*p* > 0.05). Equal uppercase letters, on the same row indicate no significant difference between blind and informed sessions for each formulation, according to paired-samples *t*-test (*p* > 0.05). FA = formulation with allulose; FS = formulation with sucrose.

## Data Availability

The original contributions presented in the study are included in the article/Appendix A, further inquiries can be directed to the corresponding authors.

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
