# Peer review of "Baru Almond Beverage (Baruccino) with Different Sweeteners: Nutritional and Physical Properties and Exploration of Sensory and Non-Sensory Perceptions"

_foods, 2026, doi:10.3390/foods15010127_

Round 1

Reviewer 1 Report

Comments and Suggestions for Authors

The manuscript presents an interesting application of allulose in a baru almond-based vegetable drink, but it does not meet the experimental rigor or scientific novelty required by Foods (MDPI). The experimental design, limited to only two treatments without factorial replication or control, prevents solid statistical inferences and significantly reduces the validity of the results. The differences observed are mainly descriptive and do not allow mechanistic or causal conclusions. For the study to be reconsidered in the future, it will need to be redesigned to incorporate more treatment levels, independent biological replication, multivariate statistical analyses (factorial ANOVA, PCA, PLS), and a more in-depth scientific discussion linking the results to physicochemical and sensory fundamentals.

Author Response

The manuscript presents an interesting application of allulose in a baru almond-based vegetable drink, but it does not meet the experimental rigor or scientific novelty required by Foods (MDPI). The experimental design, limited to only two treatments without factorial replication or control, prevents solid statistical inferences and significantly reduces the validity of the results. The differences observed are mainly descriptive and do not allow mechanistic or causal conclusions. For the study to be reconsidered in the future, it will need to be redesigned to incorporate more treatment levels, independent biological replication, multivariate statistical analyses (factorial ANOVA, PCA, PLS), and a more in-depth scientific discussion linking the results to physicochemical and sensory fundamentals

R.: We acknowledge the reviewer's observation regarding the use of two formulations (sucrose vs. allulose). The experiment followed a Completely Randomized Design, with 2 treatments, in replication, but the effect of factors and their levels was not evaluated. However, the objective of this work was not to investigate the interactions between multiple factors, but to quantify the technological, physicochemical, cinetic stability, nutritional, and sensory effects of replacing sucrose with allulose in a baru-based beverage.

All analyses were conducted following methodological rigor, which can be observed by the low standard deviation of the results. Thus, although the design does not include several factors, it is adequately structured to evaluate the specific effect of sweetener substitution, which was the central hypothesis of the study. The results were analyzed using statistical tests (ANOVA, t-tests) that support valid conclusions for the proposed research questions.

Furthermore, the manuscript presents several innovative contributions not previously reported in the literature. The study of the technological properties, stability, and consumer perception regarding the application of allulose in a plant-based beverage derived from baru expands the potential applications of allulose in plant-based matrices. Moreover, this winter beverage contributes to diversification and valorization of baru almonds.

Reviewer 2 Report

Comments and Suggestions for Authors

The manuscript presents the chemical, nutritional, and sensory analyses of an innovative winter beverage prepared from baru almonds. It also investigates the replacement of sucrose with allulose. Overall, the manuscript is well organized and clearly written. Only a few minor suggestions are proposed to further improve it:

-Line 78 - Were the roasted nut powdered? If yes, please, state all the pre-treatments performed with the nuts before beverages preparation.

Line 222 - Please, add a ")" after baru almonds.

lines 239-240 - Please, re-write this sentence.

Table 2 - Please, re-write the title. How was sodium calculated? No information is given in materials and methods section.

Lines 261-262 - Do the authors mean: "contain approximately 38.20%  to 43.60% lipids ([34] consisting predominantly of unsaturated fatty acids). "?

Sections 3.4 and 3.5 - No comparison with common beverages was done. The authors are advised to compare their results with literature data.

Sections 3.6 - It would be interesting to compare  results between genders and age groups. 

Lines 253, 273, 286, 352, 363 - Include the name of the authors before the citation ([33], [34], [39], [47], [49]).

Author Response

Revisor 2

The manuscript presents the chemical, nutritional, and sensory analyses of an innovative winter beverage prepared from baru almonds. It also investigates the replacement of sucrose with allulose. Overall, the manuscript is well organized and clearly written. Only a few minor suggestions are proposed to further improve it:

-Line 78 - Were the roasted nut powdered? If yes, please, state all the pre-treatments performed with the nuts before beverages preparation.

R: We appreciate your comment. The roasted almonds were not ground. The roasted almonds were purchased from a local market. Information about the pre-treatments is described in lines 92-96.

“In the initial stage, roasted baru almonds were soaked and refrigerated (8 °C) for 15 hours to enhance beverage palatability by removing residual water-soluble tannins. Following this, the soaking water was discarded, and the nuts were rinsed. Subsequently, in the second stage, the nuts were processed with hot water, filtered, and then reprocessed with the remaining ingredients.”

Line 222 - Please, add a ")" after baru almonds.

R. We made this change. Please, see Line 230.

Lines 239-240 - Please, re-write this sentence.

R.; Thank you for your evaluation. We have rewritten this sentence. Please see Lines 252-255.

Table 2 - Please, re-write the title. How was sodium calculated? No information is given in materials and methods section.

R: Thank you for your evaluation. We have rewritten the title of Table 2 (see Lines 256-257). The determination of sodium was performed by flame photometry. We have added more information in section 2.2. Please see lines 119-121.

“For sodium determination, the beverages were digested according to the methodology described by Ferreira et al. (2021). Sodium concentrations were then measured using a flame photometer (Celm FC-280).”

Lines 261-262 - Do the authors mean: "contain approximately 38.20% to 43.60% lipids ([34] consisting predominantly of unsaturated fatty acids). "?

R: We appreciate your comment and agree that the sentence is confusing. We have rewritten the discussion (see lines 275-280).

“The total lipid content was the same for FS and FA, 5 g/100 mL.This level indicates that the beverage is low in saturated fats. Among the ingredients, baru almonds were the main contributors to the lipid fraction, containing approximately 38.20% to 43.60% of lipids, predominantly composed of unsaturated fatty acids [35].”

Sections 3.4 and 3.5 - No comparison with common beverages was done. The authors are advised to compare their results with literature data.

R: Thank you for your comment. Following your recommendation, we have included direct comparisons in Sections 3.4 and 3.5 between the results obtained and published data for plant-based beverages or similar beverages available in the literature. There are few studies in the literature on the physical stability of beverages using the Turbiscan Lab Soft. In general, the characterization of beverages involves less sophisticated and less robust approaches for evaluating the kinetic stability of beverages.

Section 3.4. Please, see Lines 417-421; 427-432.

“CichoÅ„ska et al. [50] used Turbiscan to evaluate the kinetic stability of bean-based beverages (non-germinated beans - BG and germinated beans - G). Beverages B and BG exhibited TSI values of approximately 0.6–0.9 and 1.3–2.0, respectively, indicating distinct stability profiles. These results further confirm the sensitivity of the technique for monitoring destabilization over time under refrigerated storage conditions.”

 “Ni et al. [51] also reported destabilization phenomena when assessing the physical stability of a cloudy ginkgo beverage, despite the incorporation of a combination of thickeners (microcrystalline cellulose and gellan gum) at different concentrations. Turbiscan analysis revealed irreversible phase separation, which may have been induced by particle size effects and by the reduction in viscosity during heat treatment.”

 Section 3.5. Please, see Lines 498-505.

 “These results are consistent with the values previously reported and described in the scientific literature for plant-based beverages made with almonds and other sources. Di Renzo [56], when microbiologically evaluating different plant-based beverages (almond, Brazil nut, macadamia, cashew nut, pistachio and oat), reported that, in general, the for-mulations presented microbial counts within the limits required by legislation. These au-thors concluded that microbiological quality is a critical factor in ensuring the safety of these beverages and is related to the intrinsic composition of the product and the level of rigor adopted throughout the processing.”

Sections 3.6 - It would be interesting to compare results between genders and age groups. 

R.: We appreciate the suggestion. Indeed, analyses by gender and age group could broaden the understanding of the acceptance profile. However, this type of segmentation was not part of the original scope of the study, which was designed to assess the overall perception of regular consumers of winter beverages. Furthermore, the sample, while numerically adequate for sensory testing, was not structured to allow for rigorous statistical analyses between demographic groups.

Lines 253, 273, 286, 352, 363 - Include the name of the authors before the citation ([33], [34], [39], [47], [49]).

R.: Thank you for your evaluation. The names of the authors before the citation were included in the text.

Reviewer 3 Report

Comments and Suggestions for Authors

In this manuscript, the authors showed that baruccino is an health beverage. This winter beverage contributes to diversification and valorization of baru almonds. The study demonstrates the replacement of sucrose with allulose no modified the centesimal and chemical composition, rheological characteristics.

The manuscript is well organized and written, the experiments are well-designed, but the results should be clarified. I recommend Reconsider after major revisions.

1. Material and Methods:

Line 93: “slow pasteurization,”. To clarify the process, I suggest adding the choice of slow pasteurization.

2. Results and discussion

Line 232-233: “Paired-sample t-test”. The Table 2 no was applied statistical analysis. Table 3 and Table 4 was used Tukey test, but the correct is t-test. I suggest applying the one-way analysis of variance (ANOVA) with t-test (to compare the average values using a significant difference (p < 0.05)).

Table 6 and Table 7 are not correct the statistical analyses. Which was statistical analysis applied?. I suggest applying the one-way analysis of variance (ANOVA) with t-test (to compare the average values using a significant difference (p < 0.05)).

Line 283: Why the satured fat no were quantified in FS and FA formulations? The authors cite in work the importance this approach. I suggest adding this analysis.

Author Response

Revisor 3.

In this manuscript, the authors showed that baruccino is an health beverage. This winter beverage contributes to diversification and valorization of baru almonds. The study demonstrates the replacement of sucrose with allulose no modified the centesimal and chemical composition, rheological characteristics.

The manuscript is well organized and written, the experiments are well-designed, but the results should be clarified. I recommend Reconsider after major revisions.

  1. Material and Methods:

Line 93: “slow pasteurization,”. To clarify the process, I suggest adding the choice of slow pasteurization.

R.: Thank you for your evaluation. We have added the time and temperature for slow pasteurization. Please see line 97.

  1. Results and discussion

Line 232-233: “Paired-sample t-test”. The Table 2 no was applied statistical analysis. Table 3 and Table 4 was used Tukey test, but the correct is t-test. I suggest applying the one-way analysis of variance (ANOVA) with t-test (to compare the average values using a significant difference (p < 0.05).

R.: Thank you for your observation. We have corrected the statistical methods section and the table legends. The data in Table 2 were subjected to one-way ANOVA (α = 0.05). For comparisons between the two formulations (FS vs FA; Tables 3 and 4), we used one-way ANOVA; since there were only two groups, a post-hoc test (Tukey) was not applied. One-way ANOVA and the t-test for independent samples are equivalent, as both test whether the means differ. Comparisons between Session 1 (blind) and Session 2 (informed) were performed using a paired t-test, as they involved repeated measures on the same evaluators. Please, see section 2.8 (lines 241-248), Table 2, Table 3 and Table 4.

Table 6 and Table 7 are not correct the statistical analyses. Which was statistical analysis applied?. I suggest applying the one-way analysis of variance (ANOVA) with t-test (to compare the average values using a significant difference (p < 0.05)).

R.: We appreciate the observation and clarify that the statistical descriptions in the captions and in the Methods section have been corrected to accurately reflect the tests applied. For Tables 6 and 7, which present the sensory results, we applied the following procedures:

- Comparison between FS and FA within each session (blind or informed): a one-way ANOVA was used, equivalent to the t-test for two means, since there are only two treatments.

- Comparison between sessions 1 and 2 (blind vs. informed) for each formulation: a paired t-test was used, given that the evaluations were performed by the same consumers in both sessions.

Line 283: Why the saturated fat no were quantified in FS and FA formulations? The authors cite in work the importance this approach. I suggest adding this analysis.

R.: We appreciate your suggestion. However, the characterization of the formulations was carried out to quantify total lipids in order to construct the centesimal composition table of the formulations, which was presented as part of the sensory evaluation. Unsaturated fats are naturally present in baru almonds, predominantly in the form of monounsaturated fatty acids, and this information is well established in the literature, where several authors have reported both their occurrence and associated benefits (Santos et al., 2024; Alves-Santos et al., 2021; Oliveira-Alves et al. 2020). Therefore, we believe that the analysis of saturated fat is not essential for our work since there is no difference between FS and FA (both have the same lipid source: baru); sucrose and allulose do not influence the content or profile of fatty acids; the literature already extensively characterizes the fat profile of baru, and adding this analysis would not modify the study's conclusions.

Santos et al., 2024:  Baru (Dipteryx alata): a comprehensive review of its nutritional value, functional foods, chemical composition, ethnopharmacology, pharmacological activities and benefits for human health.  https://doi.org/10.1590/1519-6984.278932

Alves-Santos et al. 2021: Baru (Dipteryx alata Vog.) fruit as an option of nut and pulp with advantageous nutritional and functional properties: A comprehensive review. https://doi.org/10.1016/j.nfs.2021.07.001

Oliveira-Alves et al. 2020:  Identification of functional compounds in baru (Dipteryx alata Vog.) nuts: Nutritional value, volatile and phenolic composition, antioxidant activity and antiproliferative effect. 10.1016/j.foodres.2020.109026

Round 2

Reviewer 1 Report

Comments and Suggestions for Authors

The authors’ response does not address the fundamental methodological weaknesses. The study remains limited to two formulations, lacks a factorial design, and lacks independent biological replication, which prevents solid conclusions. The results are primarily descriptive and lack sufficient scientific novelty for Foods (MDPI). I therefore maintain my recommendation for rejection.

Author Response

Dear Reviewer.

We appreciate your feedback and have revised the document to ensure our reflection is fully reflected in the manuscript. Specifically, we reinforced the rationale for the scope and novelty, clearly outlining the scientific relevance of the investigation of plant-based winter beverages formulated with baru almonds and the technological implications of replacing sucrose with allulose. Please see lines 68-86 (Introduction) and 651-654 (Conclusion).

Introduction:

“Despite these advancements, this study represents a scientific contribution regarding plant-based winter beverages developed with baru almonds, particularly with respect to sugar substitution using allulose and the influence of this information disclosure on sensory perception. In this sense, the objective was to evaluate how label information on these beverages affects sweetness perception when sucrose is replaced with allulose, using consumer sensory evaluation under blind and informed conditions. In addition, physicochemical and technological characterizations were conducted, including proximate composition, pH, color, soluble solids content, microbiology, rheology behavior, and kinetic destabilization analyses. In this regard, the research has the potential to provide valuable insights for the food and beverage industry as it seeks to meet the growing demand for healthier and more sustainable products aligned with consumer expectations.”

Conclusion:

“The replacement of sucrose with allulose can be carried out without modifying the proximate composition, the physicochemical composition, rheological behavior, and kinetic stability of the developed plant-based beverages, which expands the potential application of allulose in plant matrices.”

Regarding the experimental design, we describe it in more detail in Statistical Analysis (section 2.8).

“The experiment was conducted using a completely randomized design with two formulations one with allulose (FA) and the other with sucrose (FS), in two replications. The results of the proximate, physical, and technological characterizations were evalu-ated using one-way ANOVA.

Sensory data were evaluated in two ways. For the comparison between FS and FA in each session (blind or informed), an analysis of variance (ANOVA) was performed, with two sources of variation (treatments and assessor) for each attribute. For the comparison between the blind and informed sessions for each formulation, paired-sample t-tests were performed to evaluate the influence of information on consumers' perception of attributes in the RATA and on sensory acceptability.

The results are expressed as mean ± standard deviation. Statistical analysis was performed using R software, and the significance level adopted was α=0.05.”

Reviewer 3 Report

Comments and Suggestions for Authors

In this manuscript, the authors showed that baruccino (beverage developed with baru nuts) contributes to the diversification and valorization of this Brazilian fruit. The study demonstrates the replacement of sucrose with allulose no modified the centesimal and chemical composition, rheological characteristics of baruccino. This study concludes that allulose emerges as a potential alternative to replace sucrose in plant-based beverages, appealing catering to consumers seeking pursuing well-being and health.

The manuscript is well organized and written, the experiments are well-designed, and the results were clarified. I accept in present form.

Author Response

Dear Reviewer,

We appreciate your comments and suggestions; they improved the quality of the manuscript.